# Recent Advances in AIV Biosensors Composed of Nanobio Hybrid Material

**DOI:** 10.3390/mi9120651

**Published:** 2018-12-09

**Authors:** Taek Lee, Jae-Hyuk Ahn, Sun Yong Park, Ga-Hyeon Kim, Jeonghyun Kim, Tae-Hyung Kim, Inho Nam, Chulhwan Park, Min-Ho Lee

**Affiliations:** 1Department of Chemical Engineering, Kwangwoon University, Seoul 01899, Korea; tnsdyd5015@naver.com (S.Y.P.); 1497rg@hanmail.net (G.-H.K.); 2Department of Electronic Engineering, Kwangwoon University, Seoul 01899, Korea; jaehahn@kw.ac.kr; 3Department of Electronics Convergence Engineering, Kwangwoon University, Seoul 01899, Korea; jkim@kw.ac.kr; 4School of Integrative Engineering, Chung-Ang University, Seoul 06974, Korea; thkim@cau.ac.kr; 5Division of Chemistry & Bio-Environmental Sciences, Seoul Women’s University, Seoul 01797, Korea; inhonam@swu.ac.kr

**Keywords:** avian influenza virus biosensor, avian influenza virus, electrochemical detection, electrical detection, localized surface plasmon resonance, fluorescence, nanobio hybrid materials

## Abstract

Since the beginning of the 2000s, globalization has accelerated because of the development of transportation systems that allow for human and material exchanges throughout the world. However, this globalization has brought with it the rise of various pathogenic viral agents, such as Middle East respiratory syndrome coronavirus (MERS-CoV), severe acute respiratory syndrome coronavirus (SARS-CoV), Zika virus, and Dengue virus. In particular, avian influenza virus (AIV) is highly infectious and causes economic, health, ethnical, and social problems to human beings, which has necessitated the development of an ultrasensitive and selective rapid-detection system of AIV. To prevent the damage associated with the spread of AIV, early detection and adequate treatment of AIV is key. There are traditional techniques that have been used to detect AIV in chickens, ducks, humans, and other living organisms. However, the development of a technique that allows for the more rapid diagnosis of AIV is still necessary. To achieve this goal, the present article reviews the use of an AIV biosensor employing nanobio hybrid materials to enhance the sensitivity and selectivity of the technique while also reducing the detection time and high-throughput process time. This review mainly focused on four techniques: the electrochemical detection system, electrical detection method, optical detection methods based on localized surface plasmon resonance, and fluorescence.

## 1. Introduction

Viruses are unique organisms that have various effects on living things, such as microorganisms, plants, animals, and human beings [1,2]. Since the beginning of the history of living things, viruses and living things have affected each other, leading to the evolution of advanced organisms, such as the immune system and population [3]. Nowadays, various local virus mutants are arising because of globalization and the development of worldwide transportation systems [4,5]. Humans can visit anywhere in the world very conveniently by airplane and can be easily infected by endemic viruses. When the infected human returns to their home country, the endemic virus from another place can easily settle down in their home and spread to new hosts [6]. Influenza virus is one of the general viruses in human life that can be easily infected from human to human [7]. The influenza virus can be classified with four types (A, B, C and D) corresponding to complement fixation test. The influenza virus can be divided by hemgaglutinin proteins (H number) and neuraminidase proteins (N number). So far, 18 types of H proteins and 11 types of N proteins have been reported [8]. Depending on the origin host and H-N numbers, influenza A viruses can be classified as human influenza (H1N1, H1N2, H2N2, H3N2 etc.) avian influenza (H1N1, H1N8, H2N9, H5N1 etc.), swine influenza (H1N1, H1N2, H2N1, H3N1, H3N2, and H2N3 etc.) or other types of animal influenza viruses. The structural difference between avian, human, and swine is originated from conformational change of hemagglutinin (HA) proteins [9,10].

Fortunately, influenza viruses from human to human are relatively controllable, compared to those spread by animals such as rats, deer, or birds. It is very easy to identify, isolate, and cure a human host in a hospital. However, if birds are the host of an endemic virus, the situation becomes serious and uncontrollable, because it is impossible to exactly track the destinations of every flock of birds, and they can cross borders freely [11].

Avian influenza virus (AIV) is one of the influenza viruses. AIV is classified with the negative-sense single-strand ribonucleic acid (ssRNA) virus that had the hemgaglutinin (H) number and neuraminidase (N) number combination for the virus sub-types. AIV can infect not only birds but also other animals such as swine, cat, dog, and even human. In particular, AIV H5N1 is the bird-oriented virus with high infection rates and serious symptoms such as cough, fever, and chill, which lead to death in both poultry and humans. Since the AIV H5N1 was first reported in China in 1996, the global spread of AIV H5N1 has continued with increasing economic and social damages to humans and poultry [12]. As a result, the significance of rapid, sensitive, and accurate AIV diagnosis is regarded as a prerequisite to treatment and vaccine development [13]. Viral isolation [14], immune chromogenic assay (ICA) [15], enzyme-linked immunosorbent assays (ELISA) [16], and polymerase chain reaction (PCR) and reverse transcription (RT)-PCR-based nucleic acid amplification techniques are the conventional techniques that are currently commercially used in the market [17,18]. However, viral isolation took a long time for rapid detection, ICA shows low sensitivity, ELISA requires multiple steps and is complicated to use, and PCR only provides virus detection with no genetic information.

Since 2000, various kinds of nanobiotechnology-based AIV detection methods have been reported as new alternatives. The combined usage of a nanobio hybrid material can provide various options for improving the sensitivity and selectivity and reducing the measurement time and manufacturing cost [19,20,21,22]. Nanobiotechnology has exhibited new possibilities for detecting AIV based on the unique properties of nanobio hybrid materials (Figure 1). Several reviews discussed the conventional influenza virus detection methods [13], aptasensor-based virus detection [23], rapid detection of influenza infections [24], magnetic nanobeads-based biosensor [25] and electrochemical biosensors for influenza virus detection [26,27]. To our best knowledge, there is no study focusing on AIV detection using nanobio hybrid materials. The present review discusses the recently developed AIV detection systems using nanobio hybrid materials, with a main focus on the electrochemical detection method, the electrical detection method, the surface plasmon resonance-based detection method, and the fluorescence-based detection method.

## 2. Electrochemical (EC) Detection of Avian Influenza Virus (AIV) Using Nanobio Hybrid Materials

The electrochemical (EC) detection method has various advantages: (1) rapid detection time, (2) simple apparatus, (3) easy to handle, and (4) portable fabrication for rapid virus detection in the field [25,26]. The general working principle of EC-based detection is redox reactions of electroactive species. In the case of viral protein or gene detection, the target does not have the redox species required for additional electrochemical label agents such as methylene blue, hemin or redox enzyme horseradish peroxidase, alkaline phosphatase. The EC-based biosensor can be classified as employing voltammetric, amperometric, and impedimetric techniques [27]. The change of potential, current, and impedance corresponding to redox property provides cues for binding events between the receptor and target. For virus detection, the EC-based biosensor can be divided into three groups based on bio probe type, (1) nucleic acid-modified electrode (genosensor) [28], (2) antibody-modified electrode (immunosensor) [29], and (3) aptamer-modified electrode (aptasensor) [30]. However, the EC-based biosensor only provides indirect information between the additional redox label agent and target and also has relatively low sensitivity compared to the other techniques. To overcome this problem, the nanobio hybrid materials were introduced to the EC biosensor for AIV detection. Usually, the nanobio hybrid materials such as gold nanoparticles [31], graphene [32], magnetic nanoparticle [33] etc. are combined with antibody or aptamer used to construct electrode surface fabrication for EC signal enhancement. This nanobio hybrid provides high sensitivity compared to the general metal substrate.

Zhu et al. developed a label-free biosensor composed of a DNA probe and a multi-walled carbon nanotubes-cobalt phthalocyanine–poly (amidoamine) (MWCNTs-CoPC/PAMAM) nanocomposite modified glassy carbon electrode for detecting AIV genotype [34]. They reported that the MWCNTs-CoPC/PAMAM hybrid material enhanced the guanine oxidation signal during the DNA hybridization process. This study used the guanine oxidation at +0.82 V as the redox signal measurements through differential pulse voltammetry (DPV) measurement. The DNA hybridization event between target DNA and bio probe DNA showed different oxidation current values (ΔIp) corresponding to fully matched and mismatch, which gave the selectivity. They reported that the DNA on MWCNTs-CoPC/PAMAM-modified glassy carbon electrode could determine a single mismatch of the target DNA. In addition, the fabricated sensor exhibited a limit of detection (LOD) around 0.01 ng/mL of analytical performance. Interestingly, the graphene oxide-H5-polychonal antibodies-bovine serum albumin (GO-PAb-BSA) nanobio hybrid material can be applied to develop the EC immunosensor for AIV detection [35]. This study employed the sandwich-type detection method for H5 antigen detection. The monoclonal H5 antibody as the bio probe was immobilized onto Au electrode in order to capture the H5 antigen. Then, the GO-PAb-BSA nanobio hybrid material was added to the H5 antigen-added electrode as the EC signal amplification agent. The ferricyanide was used as the redox reporter. Electrochemical impedance spectroscopy (EIS), cyclic voltammetry (CV), and DPV were employed to confirm the EC signal amplification effect of GO-pAb-BSA. When the GO-pAb-BSA was introduced, The LOD was determined to be around 2–15 to 2–8 HA unit/50 μL. Without GO-pAb-BSA introduction, the LOD was determined to be around 2–7 to 20 HA unit/50 μL. This study revealed the possibility of increasing the LOD through the introduction of nanobio hybrid material.

Notably, Veerapandian et al. reported the ultra-fast dual-target immunosensor composed of monoclonal H5N1 antibody and H1N1 antibody on methylene blue (MB)-GO-modified carbon-based screen-printed electrode (CSPE) via chitosan and protein A, respectively [32]. The fabricated dual-biosensor can detect the H5N1 and H1N1 antigen specifically using DPV and chronoamperometry (CA) technique. The detection time takes 1 min for determining the HA antigen species. For triggering electrochemical reaction, the MB was adsorbed to GO to provide the enhanced redox property between HA protein and antibodies. Figure 2A shows the fabrication process of dual immunosensor, two working electrodes were composed of graphene oxide-methylene blue/circumsporozoite (GO-MB/CS)/protein-A/antibody electrodes for detecting H5N1 and H1N1, respectively. Figure 2B describes the DPV results of detection of various H5N1 concentrations (0 pM, 25 pM, 50 pM, 75 pM, 100 pM and 500 pM). Figure 2C shows the histogram plot of selected relevant anodic peaks. These results show the fabricated immunosensor showing the linearity corresponding to the H5N1 concentrations. Figure 2D depicts the linear fitting data points that provided the LOD is 8.3 pM. This biosensor can detect the 9.4 pM of H1N1 antigen, respectively. Similar to this, the EC-based AIV detection platform can rapidly detect a target for field-ready biosensor fabrication. Also, gold nanoparticles [36] and CNT/MoSx [37] hybrid material were applied to AIV biosensor fabrication. Recently, specific proteins expressed by influenza viruses such as PA-X and PB1-F2 were reported [38,39]. Those proteins can be applied to cues for detecting AIV accurately. Miodek et al. fabricated the EC-based biosensor for PB1-F2 in infected cells [40]. Thus, EC-based AIV detection method with nanobio hybrid material shows several advantages. However, a labeling process and detection current is diminished by AIV degradation during the measurement and should be solved. Also, hybrid material is needed for optimal conditions for conjugation for AIV detection. Nevertheless, when the nanobio hybrid material is introduced properly, the EC-based AIV detection system can be solved to overcome those limitations well.

## 3. Field-Effect Transistor (FET)-Based Electrical Detection of AIV Using Nanobio Hybrid Materials

The electrical detection of biomolecules including AIVs with field-effect transistors (FETs) provides several advantages: high sensitivity in the pico- and femtomolar range, ease of use based on the label-free technique, portability due to the compact sizes of sensors and readout circuits, and low-cost mass production using a commercial semiconductor process [41,42,43,44]. The detection mechanism of AIVs using FETs is based on the direct measurement of a change in the electrical characteristics of the FET caused by a field effect from the binding of target biomolecules (i.e., nucleic acids, nucleoproteins, antigens, and antibodies) on the semiconductor channel. Recently, nanobio hybrid materials have been applied for the fabrication of the FET-based AIV biosensors, enhancing the sensitivity and reducing the detection time. Gu et al. demonstrated a nanogap FET for the electrical detection of AIV H5N1 [45]. A 20-nm-thick nanogap underneath the bridge-like suspended gate was formed on top of the channel between the source and drain. To capture a specific antibody against AI (anti-AI), they developed a silica-binding protein (SBP) fused with AI antigen which served as a bio receptor binding to the SiO_2_ surface strongly without any surface modification. The antigen–antibody binding inside the nanogap increases the dielectric constant of the nanogap, and consequently the threshold voltage decreases. The LOD of the nanogap FET biosensor was 50 ng/mL. Kim et al. improved the LOD for the detection of anti-AI down to 1 ng/mL using a charge-pumping method, which extracts a change in the interface trap density of a FET [46]. This high sensitivity is attributed to the modulation of the energy level of the trap caused by the anti-AI molecules bound in the nanogap.

Lin et al. developed a Si nanowire FET for the detection of high pathogenic strain virus (H5 and H7) DNA of AI, down to an LOD of 1 fM [47]. The ultrahigh sensitive detection of DNA is attributed to the high surface-to-volume ratio of Si nanowires, resulting in a high current change due to a field effect. Poly-crystalline Si nanowires were fabricated through the sidewall spacer technique, which enables the definition of the nanosized patterns without needing to employ an expensive lithography process. AIV H5- and H7-captured DNA probes were immobilized on the Si nanowires. The increased negative charges due to hybridization between target DNA and the captured DNA probes resulted in a current change of the Si nanowire FET biosensor. They confirmed the selectivity of the sensor with control experiments in which the transfer curves of the unmodified Si nanowire FETs remained unchanged in PBS buffer as well as in the presence of H5 and H7 target DNA. Ahn et al. developed a silicon nanowire FET-based biosensor to detect anti-AI in real-time [48]. The drain current of the FET changed upon injection of an anti-AI solution onto the device functionalized with the antigen of AI. 

They also reported that a sensor signal generated by the intrinsic charges of the anti-AI can be amplified by double gates [49]. One gate is used to sweep the gate voltage for the measurement of the threshold voltage, while the other gate controls channel potential and enables the FET to be more sensitive to the charges of biomolecules. Ono et al. developed a graphene FET for the selective detection of lectins derived from sambucus sieboldiana (SSA) and maackia amurensis (MAM) as alternatives of the human and avian influenza viruses, respectively [50]. A chemical vapor deposition (CVD) grown monolayer graphene was transferred onto a Si/SiO_2_ substrate. The high mobility of graphene leads to a large current change by the biomolecular interaction on the graphene surface. Following the surface functionalization of graphene by glycan, the graphene FET biosensors show sensitive and selective detection of both SSA lectin (pseudo human-type virus) and MAM lectin (pseudo avian-type virus) with LODs of 130 pM and 150 pM, respectively. The hole current increased due to the negative surface charges of lectins. Hideshima et al. demonstrated that the glycan-immobilized FET showed the attomolar-level detection (>50 aM) of influenza A viral HA molecules H1 and H5, which indicates a much higher sensitivity than that of the antibody-immobilized FET (>50 pM) [51]. The smaller sizes of glycans (~2 nm) compared to those of antibodies (4–12 nm) allows for sensitive detection based on FET sensors because of the densely immobilized receptors and the proximity of target biomolecules to the sensor.

Chan et al. developed a reduced graphene oxide (rGO) FET for H5N1 influenza virus gene detection [52]. Rather than the CVD growth and transfer method, GO flakes suspended in the solution were coated onto a Si/SiO_2_ substrate as a film by spin-coating, then hydrazine reduction was performed in order to obtain rGO film (Figure 3A). In a flowing environment, conventional short capture probes are washed away from the rGO surface after hybridization with target. However, extended long-capture probes are kept on the rGO surface due to the immobilization section, which can result in higher LOD. The drain current is reduced after probe immobilization and hybridization with complementary target DNA (cDNA) due to negative charges of DNA (Figure 3B). The selectivity of the rGO FET biosensor is confirmed from negligible change in the drain current after binding with non-complementary target DNA (non-cDNA) as shown in Figure 3C. The rGO FET biosensor working in a flowing environment with an extended long-capture probe showed an LOD of 5 pM with a detection time of 1 h (Figure 3D,E). The rGO FET biosensor working in a flowing environment with an extended long-capture probe showed an LOD of 5 pM with a detection time of 1 h. Park et al. reported the on-site diagnosis of AIV infection by a FET biosensor that detects H9N2 virus nucleoproteins within 30 min, down to an LOD of 10^3^ EID_50_/mL using cloacal swabs acquired from a live chicken [53]. A Si FET, as a transducer, was connected to a disposable well gate (DWG) that is easily replaced after measurement. Nucleoproteins released from the virus in the lysis buffer were selectively bound to the AIV antibodies conjugated on the DWG surface. The intrinsic charges of nucleoproteins result in changes in the threshold voltage of the FET transducer by a field effect. Compared to commercial optical AIV rapid kits, the LOD of the proposed FET-based AIV sensor was improved by at least one order of magnitude.

Although FET biosensors combined with nanobio hybrid materials allow for the rapid, label-free electrical detection of AIVs with very low LOD, the detection signal and sensitivity are degraded in high ionic concentrations by the Debye screening effect, which make it difficult to use FET biosensors in practical applications. Further studies are required for AIV detection to overcome the Debye screening effect, with a focus on techniques such as buffer exchange, receptor engineering, and the signal amplification method. Table 1 shows the comparison of EC-based biosensor and FET-based for AIV detection in terms of several factors.

## 4. Surface Plasmon Resonance (SPR)-Based Detection of AIV Using Nanobio Hybrid Materials

Over the last three decades, the surface plasmon resonance (SPR) technique has been applied to detect the nanoscale interface change that provides various applications for biosensor applications. Because of their portability and high sensitivity, SPR-based biosensors were regarded as a good solution to the problem of constructing a field-ready pathogen detection tool [54,55,56]. When the light was exposed to the surface of the metal substrate, the electrons oscillated resonantly between the electron and charged metal surface that produced the surface plasmon effect. These surface changes can be used as cues for the interface interaction between the target molecule and the bio probe-modified substrate, which is usually a noble metal or nanoparticle [57]. In particular, SPR detection methods can be easily tuned for sensitivity, selectivity, and detection time with the nanomaterials on the substrate. Recent advances in nanobiotechnology provide various chances to develop the SPR-based biosensor for detecting biological targets [58,59]. This novel technique was clearly easily applied to AIV detection [60,61,62,63]. Usually, SPR-based biosensors were developed for one of a few categories: (1) HA protein detection; (2) whole AIV detection; or (3) AIV gene hybridization. Usually, in the case of an SPR-based biosensor, the nanobio hybrid materials were introduced to use the localized-SPR (LSPR) effect for ultrasensitive AIV detection or to construct a functional bio probe in order to improve the analytical performance or reduce the sensor fabrication process. The bio probe such as antibody or aptamer immobilized onto the well oriented nanostructure provides a high sensitivity detection platform. Bai et al. fabricated a portable aptasensor for detecting H5N1 virus from poultry swab samples [64]. The fabricated device detected AIV concentrations from 0.128 to 12.8 HAU in 1.5 h. A SPR biosensor was also developed to detect the AI virus gene [63,64,65]. Gu’s group fabricated a H5NX aptasensor for sandwich-type SPR biosensors [66]. For developing an AIV aptamer, they introduced the multi-graphene oxide- systematic evolution of ligands by exponential enrichment (SELEX) method that provides the whole detection of AIV well. Similar to the sandwich-type ELISA, the primary AIV aptamer was immobilized on an Au substrate, then, the AIV was added. Finally, the secondary AIV aptamer was captured in order to increase the SPR signal. The LOD showed 200 EID_50_/_mL_ in this study.

Recently, Chang et al. reported the intensity modulated SPR (IM-SPR)-based immunosensor for detecting AIV H7N9 [67]. In this study, they developed a new AIV antibody using reverse genetics and this novel bio probe was used as a bio probe with IM-SPR for AIV detection. The recombinant antibody showed high sensitivity compared to the conventional antibody that can detect around 402 copies/mL in serum mimic solution. Notably, the proposed IM-SPR biosensor composed of recombinant antibody reduced the detection time by around 10 min. The analytical performances (detection time and detection limit) of the developed IM-SPR immunosensor were compared to those of antitative reverse transcription polymerase chain reaction (qRT-PCR, detection time: 2.5 h, detection limit: 498 copies/mL), target-capture ELISA (detection time: 5 h, detection limit: 2.3 × 10^4^ copies/mL), and rapid influenza diagnostic test (RIDT, detection time: 15 mins, detection limit: 402 copies/mL). Compared to the conventional AIV detection methods, the IM-SPR-based biosensor showed better sensitivity and faster detection time.

For the fabrication of the simple biosensor, LSPR is suitable as a field-ready AIV biosensor. Park et al. reported the AIV immunosensor composed of recombinant fusion protein with AIV antibody on the Au nanoparticle and Au substrate (Figure 4) [68]. Figure 4A shows the LSPR results corresponding to biosensor fabrication process. They genetically fused the gold binding peptide (GBP) to an HA antibody, which reduced the time needed for the biosensor fabrication step. For the SPRimaging (SPRi) experiment, the GBP-antibody was immobilized onto a spray-type microarray Au electrode. The LOD is 30 pg/mL using a spray-type arrayer and FL signal. For more sensitive AIV detection, they also conducted an LSPR experiment using a deposition thin Au nanofilm-encapsulated on a 60 multi-spot silica nanoparticle (Au@SNP) array chip. Then, the GBP-antibody was immobilized for HA protein recognition. The LSPR phenomena between HA protein and GBP-antibody on Au@SNP were observed at 540 nm using the gold-silica hybrid nanostructure, and the LOD was determined to be 1 pg/mL while the detection range was 1 pg/mL to 1 ug/mL using the proposed system. Figure 4B depicted the superimposed absorbance spectrum of 60 multi-spot AuNP array chip and the absorbance peak was obtained around 540 nm (Black line). When the target HA protein was added to GBP-antibody (Red line), the absorbance intensity increased (Green line). Figure 4C shows the sensitivity test of fabricated immunosensor and Figure 4D shows the calibration curves of absorbance dependence on antigen concentrations. Also, the electrically activated magnetic nanoparticle was applied to SPR for detecting pandemic influenza [69]. In this way, nanobio hybrid materials can effectively reduce the detection time and sensitivity for the SPR-based AIV biosensor. However, it is hard to perform the multiple target detection analysis and it required to highly ordered nanopattern. The future nanobio hybrid material-based AIV biosensor should consider those problems for field-ready AIV biosensor construction.

## 5. Fluorescence (FL)-Based Detection of AIV Using Nanobio Hybrid Materials

Conventionally, the fluorescence (FL)-based viral detection technique is the most common way to target a gene or viral protein [70,71]. Typically, the FL-based detection system requires a labeling process for obtaining the fluorescence signal. The working principle of FL-based detection is based on the signal on/off between the target binding to the fluorescent probe, such as dye-labeled nucleic acid, fluorescent nanoparticles, fluorescent proteins [72,73,74]. When the target was bound to probe, then the FL signal increased or decreased based on the detection strategy. Several groups have developed an FL-based AIV detection system [19,75,76,77]. Xu et al. reported a FL-based AIV H5N1 biosensor composed of aptamer-quantum dots (QDs) hydrogel [21]. This study used the QD as the fluorescence reporter and Iowa Black^®^RQ-Sp at the 3′-terminal of aptamer as a quencher to QDs for target AIV. The LOD of the fabricated hydrogel-based aptasensor is 0.4 HAU (2−1.2 to 26 HAU/ 20 uL) in 30 min. In another study, the highly luminescent CdTe/CdS QDs were synthesized for AIV H5N1 detection [78]. The H5N1 antibody conjugated with CdTe/CdS QD provided a detection limit of 3 ng/μL for H5N1 virus detection.

Recently, Ahmed et al. synthesized the molybdenum disulfide-QD (MoS_2_-QD) with magnetic nanoparticle for an FL-based AIV H5N1 detection system [79]. The MoS_2_ is the transition metal dichalcogenide (TMD) material that is also called the topological insulator (TI) (Figure 5). Usually, TIs are regarded as an interesting material beyond graphene because of their unique electrical and optical properties. Normal TI has a planar structure similar to the 2D structure of graphene. They made the 2D of MoS_2_ to a QD structure 0D that exhibited new optical properties which showed a blue emissive FL peak and circular dichroism peak at 420 nm and 330 nm, respectively. Using the chiral MoS_2_-QD (around 2–3 nm size) nanoparticle, the synthesized nanomaterial is conjugated with anti-HA antibodies to detect the AIV. In addition, the anti-neuraminidase (N) antibodies conjugated to magnetic nanoparticles (MNPs). When the target AIV H5N1 was added to MoS_2_-QD and N-MNP diluted solution, the target/MoS_2_-QD/N-MNP were formed to complex. Then, the external magnet was located at the bottom of the reaction beaker and the upper solution was transferred to new tubes that provided the photoluminescence (PL) and circular dichroism (CD) information of AIV detection (Figure 5A). Figure 5B shows the PL intensity difference after the MoS_2_-QD/N-MNP hybrid formation. Figure 5C shows the change in PL intensity corresponding to antigen concentration from 10 pg/mL to 10 μg/mL. Figure 5D depicted the calibration curve of PL response with various virus concentrations. Moreover, the selectivity test was carried out with different types of influenza (Figure 5E). The PL and CD spectra of unreacted MoS_2_-QDs give LODs of 7.35 pg/mL and 80.92 pg/mL, respectively. A few studies have reported the fabrication of AIV gene detection based on FL methods [80,81]. Recently, the label-free light-up FL detection method comprised of a DNA triplex assembly and hybridization chain reaction (HCR) amplification was proposed to detect AIV H7N9 and thrombin [82]. When the target DNA was added to the designed DNA triplex structure, the hybridization started with berberine as the FL indicator that continuously exhibited the increment of FL signal. The viral DNA was determined to be in the range of 0.2–100 nM and LOD was 0.14 nM.

Yeo et al. also proposed a field-ready FL-based AIV H5N1 detection system integrated with a smartphone [83]. They introduced the fluorescent lateral flow immunoassay with coumarin-derived dendrimer for fluorescence signal enhancement. For the clinical trial, the proposed smartphone-based FL biosensor and fluorescein isothiocyanate (FITC) assay detected various types of AIV (H5N3, H7N1 and H9N2) in patient samples. In the case of H5N3, LOD showed 6.25 × 10^3^ PFU/mL using a smartphone FL detector and 1.25 × 10^4^ PFU/mL using FICT assay, respectively. For H7N1, the LOD of the smartphone-based FL detector and FICT assays were 5.34 × 10^2^ PFU/mL and 1.06 × 10^3^ PFU/mL, respectively. In the case of H9N2, the smartphone and FICT assays showed detection limits of 5.23 × 10^1^ PFU/mL and 1.09 × 10^2^ PFU/mL, respectively. This resulting FL-based biosensor with nanobio hybrid material can be easily applied to field-ready biosensor construction. The operation time is around 15 min, which demonstrates the potential of rapid field-ready biosensor on site. This group went on to develop an epitope-derived peptide with europium nanoparticle to detect AIV H5N1 using the fluorescence-linked immunosorbent assay in the point-of-care testing (POCT) system [84]. This novel peptide can be used as the fluorescence signal reporter in the proposed detection system that can reduce the antibody size in the interface. Table 2 shows the comparison of EC-based biosensor and FET-based for AIV detection in terms of several factors. Notably, the FL-based AIV diagnosis system with the nanobio hybrid material can be easily integrated with a smartphone, which can provide the detection system with simplicity, portability, and low-cost manufacturing. However, two major concerns need more study. First, FL-based AIV detection carried out with blood sample that red color of blood hampered the FL signal. The proper pre-treatment of clinical sample is required. Also, the detection step is relatively complicated for user. The nanobio hybrid material-based AIV biosensor should consider those problems for future commercialization.

## 6. Outlook

As world globalization has accelerated with development of human and material resource exchange between nations, various virus mutants and sub-types with high pathogenicity will rapidly increase in the world. Among them, AIV is uncontrollable, serious, and infectious because of various sub-types from poultry-to-poultry infection and poultry-to-human infection. Conventional AIV detection methods including virus culture, immunochromatogenic assay and ELISA required long assay time and a trained operator that hampers the rapid-detection AIV kit development. Fortunately, the detection techniques of AIV have been developed for various types of biosensor based on nanobiotechnology. The AIV biosensor can be easily tuned for functionalities such as sensitivity, selectivity, portability, detection time, small amount of sample volume, and feasibility corresponding to adequate nanobio hybrid material. Usually, the nanobio hybrid materials were used for two purposes: (1) The fabrication of electrode substrate composed of bio probe/nanostructure that enhance the signal between target and bio probe for EC and FET-based biosensors; and (2) additional labeling agent composed of bio probe/nanoparticles for increasing the sensitivity between target and receptor for optical biosensors. Other approaches such as gravimetric viral diagnostics based on quartz crystal microbalance (QCM) can also provide rapid, label-free, and sensitive virus detection [85]. Also, a magnetic nanoparticle-based detection method could be a great tool for rapid AIV detection that can reduce the detection time and increase the sensitivity [86,87,88,89,90,91]. The giant magnetoresistive sensor is a good alternative for AIV detection [92,93]. The transducer principle is based on the change in resonant frequency of a crystal after target binding. However, they suffer from a certain inherent problem, namely mechanical instability by decreasing thickness of the quartz sheet to enhance the sensitivity and signal fluctuation in highly viscous analyte solution. Nanobio hybrid material can provide high sensitivity and stable operation at the same time in the four methods (EC, E, SPR, and FL) described in this review. It is obvious the biosensor comprised of nanobio hybrid material showed several advantages compared to conventional AIV detection methods. So far, the nanobio hybrid material-based biosensor has been developed to research level. Although those AIV biosensors required more tests and optimization processes, the introduction of nanobio hybrid to biosensors combined with the nanofabrication of electrode array integrated with a printed circuit board for miniaturization is required for the product commercialization. Also, the real-time monitoring of AIV technique should be considered to determine suspected virus infection samples [94,95,96]. In addition, nanobio hybrid material can be used in wearable biosensor platforms, which is one of the future trends in biosensors due to simple usage and long-term monitoring capability. Compared to bulk-type biosensors, nanobio hybrid material-based biosensors have superior mechanical flexibility promising in wearable and flexible applications worn on the skin. Most of the biosensors described in this review can be fabricated in wearable forms. However, new sample collection methods will be considered to obtain analyte in wearable biosensors for continuous monitoring. For example, minimally invasive microneedles, which gently extract target analyte from biofluid, can be compatible with wearable biosensors. Moreover, AIV biomarkers for non-invasive detection should be investigated further. If the AIV nanobio hybrid-based biosensor system can solve the reliability, clinical sensitivity, and feasibility issues, then the AIV nanobio hybrid material-based detection system has a chance to be the general virus detection platform integrating with a smartphone or smartwatch that will provide fast disinfection and disease control that contributes human welfare in the near future.

## Figures and Tables

**Figure 1 micromachines-09-00651-f001:**
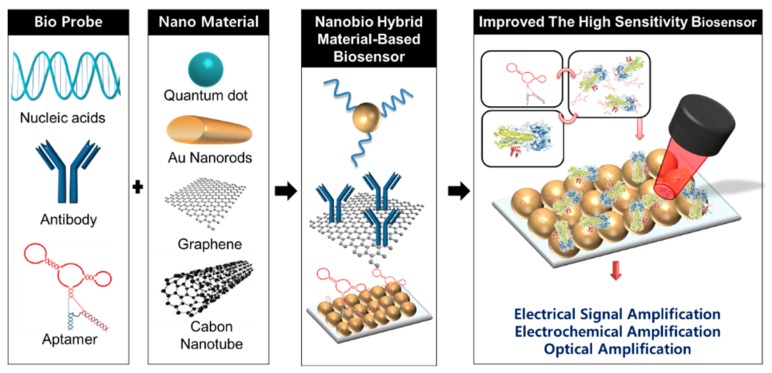
Schematic diagram shows biomaterials and nanomaterials combination allow their integration in nanobio hybrid materials for avian influenza virus (AIV) biosensor.

**Figure 2 micromachines-09-00651-f002:**
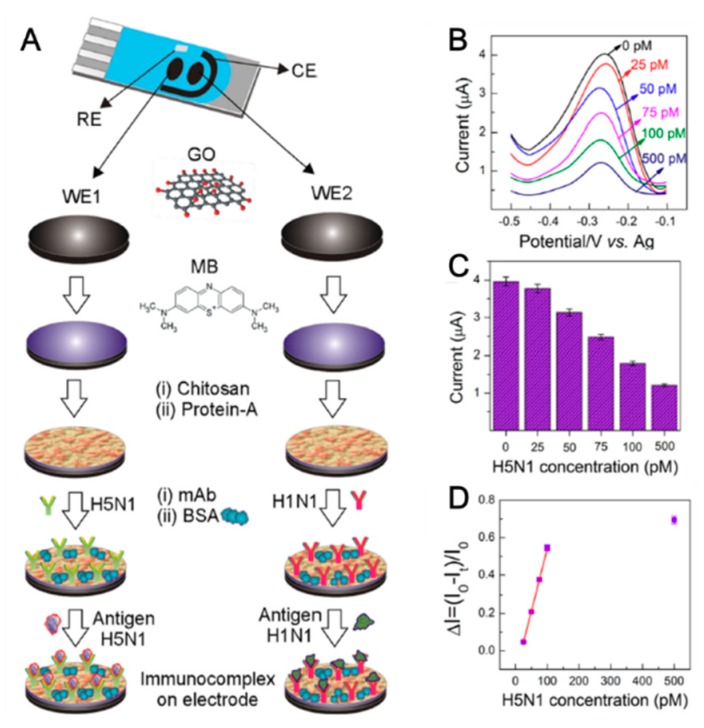
(**A**) Representation of dual screen-printed electrode (SPE) and fabrication process of immunosensor platform. WE–working electrode, CE–counter electrode, and RE–reference electrode. (**B**) Representative differential pulse voltammetry (DPVs) of graphene oxide-methylene blue/circumsporozoite (GO-MB/CS)/protein-A/anti-H1N1 electrodes against various concentrations of antigen H1N1. (**C**) The selected relevant peak voltammetric histogram. (**D**) Calibration curve derived from the DPVs expressed in ΔI. ΔI = (I0 − It/I0), where I0 is the absolute peak current value of the modified electrode in phosphate-buffered saline phosphate-buffered saline (PBS) without antigen application, and it is the peak current value of the modified electrode measured in PBS with the desired concentration of the antigen (n = 3). (Reproduced with permission from [32], published by Elsevier, 2016).

**Figure 3 micromachines-09-00651-f003:**
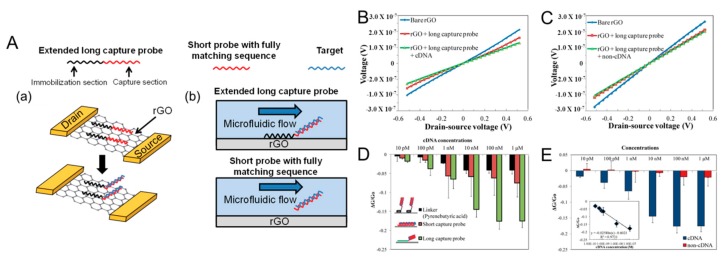
(**A**) Mechanism of extended long-capture probe immobilization strategy to keep probe stability in the flow environment. (**a**) The immobilization section of extended long-capture probe (LP) still keep π-π stacking interaction on reduced graphene oxide (rGO) surface after hybridization; (**b**) In flowing environment, short capture probes with fully match sequence after hybridization with target are washed away from rGO surface. Extended long-capture probes are still kept on rGO surface. Direct current measurement for (**B**) cDNA and (**C**) non-cDNA for bare rGO, rGO functionalized with LP and rGO functionalized with LP after target addition. (**D**) Relative conductance change (ΔG/G_0_) of rGO transistors with different probe immobilization approaches of short probes, extended long-capture probes and covalent immobilization with linkers in various cDNA concentrations in the microfluidic chip after washing steps. (**E**) Relative conductance change (ΔG/G_0_) of rGO transistors with LP at various cDNA and non-cDNA concentrations in the washing environment. Inset: linear detection range of rGO transistor using LP. (Reproduced with permission from [52], published by Elsevier, 2017).

**Figure 4 micromachines-09-00651-f004:**
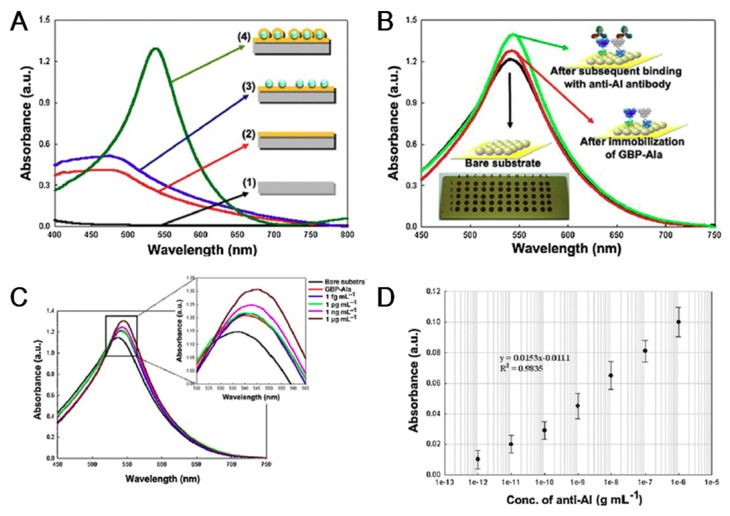
(**A**) Optical characteristics of multi-spot gold-capped nanoparticle array chip with localized-surface plasmon resonance (LSPR) absorbance peak: (1) slide glass substrate, (2) after gold deposition on slide glass substrate, (3) after silica nanoparticle array on the gold-deposited substrate, and (4) gold-capped nanoparticle array chip. Absorbance peak of multi-spot gold-capped nanoparticle array chip was observed at 540 nm. (**B**) Absorbance spectrum of LSPR properties obtained after binding of 100 μg/mL gold binding peptide (GBP)–AIa with 1 μg/mL anti-AI antibody in wavelength region (from 450 nm to 850 nm). LSPR properties of bare substrate (black line), GBP–AIa-immobilized on the multi-spot chip surface (red line), binding reaction between GBP–AIa and anti-AI antibody on the multi-spot chip surface (green line). Inset represents the 60 multi-spot gold-capped nanoparticle array chip with one spot of 2 mm in diameter. (For interpretation of the references to color in this figure legend, the reader is referred to the web version of the article). (**C**) Superimposed absorbance spectrum curves obtained from GBP–AIa immobilized on the surface of multi-spot chip in the presence of anti-AI antibody with concentrations ranging from 1 fg/mL to 1 μg/mL. (**D**) Calibration curve for absorbance dependence on anti-AI antibody concentration by using GBP–AIa-immobilized multi-spot chip. (Reproduced with permission from [68], published by Elsevier, 2012).

**Figure 5 micromachines-09-00651-f005:**
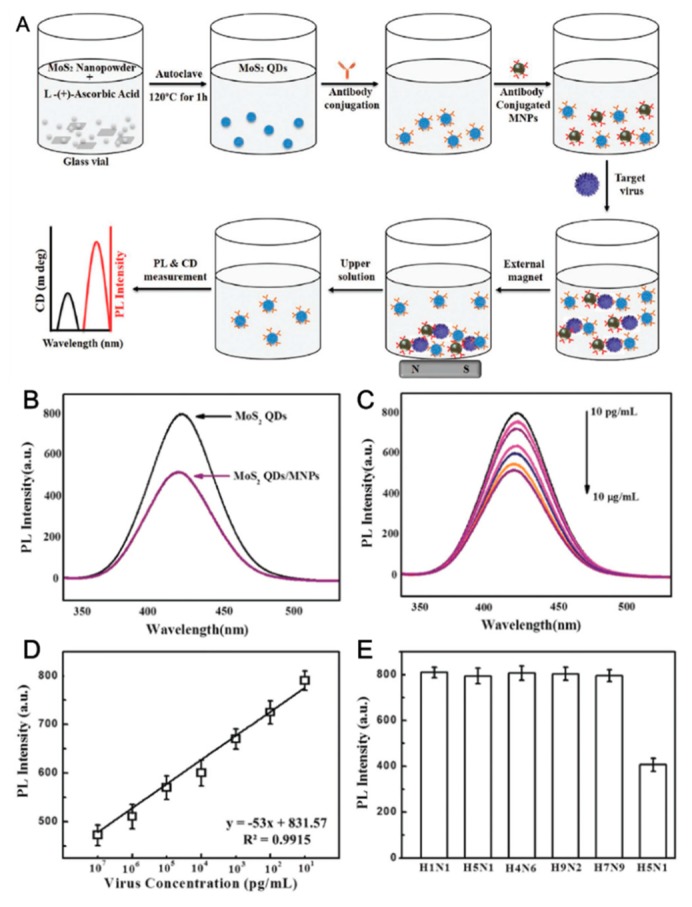
(**A**) Schematic presentation of the overall experimental design: MoS_2_ nanopowder and ascorbic acid are mixed in a glass vial; Upon autoclaving, MoS_2_ QDs were formed; aptamer-quantum dots (QDs) were conjugated with antibodies through a layer-by-layer (LBL) method; antibody-conjugated magnetic nanoparticles (MNPs) were added to it; target virus was added and nanostructured magnetochirofluorescent was formed, which were separated by an external magnet; upper solution was collected and tested for optical responses. Fluorescence detection of avian influenza virus A (H5N1): (**B**) PL response before and after nanohybrids formation; (**C**) PL spectra of QDs with different concentrated target virus solution; (**D**) calibration curve of PL intensity versus virus concentration; and (**E**) selectivity results of fluorescence-based sensor. (Reproduced with permission from [79], published by WILEY-VCH Verlag GmbH & Co. KGaA, 2018).

**Table 1 micromachines-09-00651-t001:** Comparison of the electrochemical (EC)-based biosensor and field-effect transistor (FET)-based for avian influenza virus (AIV) detection in terms of receptor materials, nanobio hybrid material, detection method, detection limit and target.

Bioreceptor	Nanobio Hybrid Material	Detection Method	Detection Limit	Target	References
Aptamer	Gold nanoparticle	EIS	0.125 HAU (pure virus)	Virus	[31]
			1 HAU (chicken swab samples)		
Antibody	Graphene	DPV	8.3 pM	HA	[32]
Antibody	ZnO Nanords	CA	1 pg/mL	HA	[33]
DNA probe	MWCNTs-CoPC/PAMAM	DPV	0.01 Ng/mL	Gene	[34]
Antibody	Graphene	DPV	2^−15^ HA unit/50 μL	HA	[35]
Antibody	4,4′-thiobisbenzenethiol/gold colloidal NPs	EIS, OSWV	0.6 pg/mL	Virus	[36]
Antibody	CNTs/MoS	LSV	0.43 ng/mL	HA (H7)	[37]
Antibody	Silica-binding protein (SBP)-fusion protein	FET	50 ng/mL	Antibody (H5N1)	[45]
DNA probe	Silicon Nanowire	FET	1 fM	Gene (H5, H7)	[47]
Glycan	Graphene	FET	130 pM	Lectin	[50]
DNA probe	rGO	FET	50 pM	Gene (H5N1)	[52]

**Table 2 micromachines-09-00651-t002:** Comparison of the localized-surface plasmon resonance (LSPR)-based biosensor and fluorescence (FL)-based for AIV detection in terms of receptor materials, nanobio hybrid material, detection method, detection limit and target.

Bioreceptor	Nanobio Hybrid Material	Detection Method	Detection Limit	Target	References
Antibody	Recombinant Antibody	IM-SPR	144 copies/mL	HA	[67]
Antibody	Goldbinding polypeptide (GBP)–fusion protein	LSPR/SPRi	1 pg/mL	HA	[68]
DNA probe	Gold nanoarray	LSPR	2.36 × 10^13^/cm^2^ oligonucleotides	Gene	[63]
Antibody	Antibody-Gold nanoparticle Antibody-QD Complex	FL-LSPR	10 pfu/mL	Virus (H3N2)	[60]
Aptamer	Aptamer-Cojugated Gold nanoparticle	SPR	200 EID_50_/mL	Virus	[66]
Aptamer	Ag@SiO_2_ nanoparticle	MEF	2 ng/mL (in aqueous buffer)	HA	[19]
			3.5 ng/mL (in Human serum)		
ntibody	CdTe/CdS	PL	3 ng/ μL	Virus (H5N1)	[77]
Antibody	MoS2 QDs	PL	7.35 pg/mL	Virus (H5N1)	[78]
DNA Probe	DNA Triplex with berberine	FL	0.14 nM	Gene	[81]
Antibody	Coumarin-derived Dendrimer	FL/FICT	H5N3: 6.25 × 10^3^ PFU/mL (FL), 1.25 × 104 PFU/mL (FICT)	Viruses (H5N3, H7N1, 9N2)	[82]
			H7N1: 5.34 × 10^2^ PFU/mL (FL), 1.06 × 10^3^ PFU/mL (FICT)		
			9N2: 5.23 × 10^1^ PFU/mL, 1.09 × 10^2^ PFU/mL		

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
