# Peer review of "Recent Advances in AIV Biosensors Composed of Nanobio Hybrid Material"

_micromachines, 2018, doi:10.3390/mi9120651_

Round 1
Reviewer 1 Report
The paper represents a review on the AIV biosensors employing nanobio hybrid materials. At the present moment I feel that the paper does not meet the requirements for publication due to the following facts:
The tile of this paper is inappropriate. This review is not focusing on the fabrication of AIV biosensors but on the recent advances in AIV biosensors. I suggest the authors to change the title to better represent the work of this paper.
Grammatical errors. I strongly suggest you to find a native English speaker to check your manuscript.
Apart from the grammatical errors, there are many writing errors in this paper. For example, line 146 detect the detect [detect]; line 305-308 [detection time and detection limit]; line 316 AI detection [AIV detection].
Put the full names of the abbreviations upon their first appearance in this paper. For example, SPR first appeared in line 252, however, the authors put the full name in line 264.
Use superscripts for the numbers. For example: lines 214, 306, etc.
All the figures in this paper are not well described in the text.
There is not proper critical insights on the state of the art not thorough thoughts on future trends, just general comments. The authors should give a deeper critical analysis.
Author Response
Comments and Suggestions for Authors
The paper represents a review on the AIV biosensors employing nanobio hybrid materials. At the present moment I feel that the paper does not meet the requirements for publication due to the following facts:
The tile of this paper is inappropriate. This review is not focusing on the fabrication of AIV biosensors but on the recent advances in AIV biosensors. I suggest the authors to change the title to better represent the work of this paper.
Authors agreed with reviewer’s suggestion, the title is changed to ‘Recent advances in AIV biosensors composed of nanobio hybrid material’
Grammatical errors. I strongly suggest you to find a native English speaker to check your manuscript.
Authors agreed with reviewer’s suggestion, we checked the manuscript carefully for appropriate corrections, English language style, grammar and spelling with the red color. Also, please find the certificate for English-editing of enclosed manuscript.
Apart from the grammatical errors, there are many writing errors in this paper. For example, line 146 detect the detect [detect]; line 305-308 [detection time and detection limit]; line 316 AI detection [AIV detection].
Authors agreed with reviewer’s suggestion, we checked the manuscript carefully for appropriate corrections with the red color.
Put the full names of the abbreviations upon their first appearance in this paper. For example, SPR first appeared in line 252, however, the authors put the full name in line 264.
Authors agreed with reviewer’s suggestion, we modified the mistake with the red color. Thanks for reviewer’s comment .
Use superscripts for the numbers. For example: lines 214, 306, etc.
Authors agreed with reviewer’s suggestion, we modified the mistake with the red color.
All the figures in this paper are not well described in the text.
Authors agreed with reviewer’s suggestion, we explained the figures more clearly. Please check the description of Figure 2, 3, 4, 5 , respectively.
There is not proper critical insights on the state of the art not thorough thoughts on future trends, just general comments. The authors should give a deeper critical analysis.
Authors agreed with reviewer’s suggestion, please check the outlook session .
Reviewer 2 Report
Comments to the Author
The manuscript “Fabrication of Avian Influenza Virus Biosensor using Nanobio Hybrid Materials” by Lee et al., presents an extensive body of literature data on development of new detection methods namely electrochemical, electrical, LSPR and fluorescence for influenza viruses based on nanobio materials.
I can recommended this review for publication in the Micromachines after minor modification.
I propose several modifications:
1) In Introduction-
2nd paragraph , p.2 Precise what are avian influenza viruses. Influenza virus are classified as type A, B, C,D. Avian viruses belong to the influenza virus type A. The differences between human, avian, and swine viruses should be added to strass what is the particularity in the detection of avian virus biomarkers.
2) In the main text-
Add some previous reviews dealing with advanced methods for influenza virus detection in poultry and veterinary in general. This is needed in order to stress the novelty of the present manuscript compared to previous reviews in this fields.
3)
Recently some new proteins were found to be expressed by influenza viruses like PA-X, and PB1-F2. Especially PB1-F2 is a biomarker of avian virus strains. This small protein was shown to be involved in pathogenicity of the avian viruses and there are some electrochemical biosensors already developed for PB1-F2 detection.
Those biosensors should be added into the review, as such new biosensors help not only detection of the avian virus biomarkers but also the fundamental research in influenza virus field
Author Response
I can recommended this review for publication in the Micromachines after minor modification.
I propose several modifications:
1) In Introduction-
2nd paragraph , p.2 Precise what are avian influenza viruses. Influenza virus are classified as type A, B, C,D. Avian viruses belong to the influenza virus type A. The differences between human, avian, and swine viruses should be added to strass what is the particularity in the detection of avian virus biomarkers.
Authors agreed with reviewer’s suggestion. We have added the description of influenza virus type and difference between avian, human and swine influenza viruses. Depending on the origin host and H-N numbers, influenza A viruses can be classified as human influenza (H1N1, H1N2, H2N2, H3N2 and etc.) avian influenza (H1N1, H1N8, H2N9, H5N1 and etc.), swine influenza (H1N1, H1N2, H2N1, H3N1, H3N2, and H2N3 and etc.) or other types of animal influenza viruses. In case of same subtype (Bird to human case or bird to swine case). It is little structural difference between avian, swine and human influenza virus. Only the difference is conformation of HA protein. Cueno et al reported the structural differences between the avian and human H7N9 HA proteins. Also, Baumann et al studied the structural regions of HA2 subunit induced the conformational transition of HA between avian influenza and swine influenza. To determine the origin and species exactly, we have to construct bioprobe such as antibody or aptamer that should be confirmed with high selectivity and cross activity. And, PCR can be provided the genetic information of influenza virus that help the strain identification. We also referred those two additional references in the manuscript. Please check the references 9,10
Marni E. Cueno, Kenichi Imai, Muneaki Tamura, Kuniyasu Ochiai, Structural Differences between the Avian and Human H7N9 Hemagglutinin Proteins Are Attributable to Modifications in Salt Bridge Formation: A Computational Study with Implications in Viral Evolution, PLOS ONE, 8, 2013, e76764
Jan Baumann, Nancy Mounogou Kouassi, Emanuela Foni, Hans-Dieter Klenk, Mikhail Matrosovich, H1N1 Swine Influenza Viruses Differ from Avian Precursors by a Higher pH Optimum of Membrane Fusion, J. Virol. 90, 2016, 1569
2) In the main text-
Add some previous reviews dealing with advanced methods for influenza virus detection in poultry and veterinary in general. This is needed in order to stress the novelty of the present manuscript compared to previous reviews in this fields.
Authors agreed with reviewer’s suggestion, we have added the additional reviews about AIV detection in poultry and veterinary in general. Thanks for reviewer’s comment that makes the manuscript better. Please check the references 23-26.
3)
Recently some new proteins were found to be expressed by influenza viruses like PA-X, and PB1-F2. Especially PB1-F2 is a biomarker of avian virus strains. This small protein was shown to be involved in pathogenicity of the avian viruses and there are some electrochemical biosensors already developed for PB1-F2 detection. Those biosensors should be added into the review, as such new biosensors help not only detection of the avian virus biomarkers but also the fundamental research in influenza virus field
Authors agreed with reviewer’s suggestion, we have added the references considering PB1-F2 detection and PA-X. Please check references 38,39,40.
Reviewer 3 Report
The manuscript presents a comprehensive and systematic study on a number of biosensors for avian influenca virus based on nanobio hybrid materials. The text is clearly written. The style is fine, but minor spell check is required. This relates to the lines 36, 60, 61, 70, 103, 121, 134, 140, 150, 153, 195, 209, 214, 229, 232, 277, 299, 306, 315, 321, 336, and 338. Maybe few more.
In addition, at the end of the manuscript, a short view on competing transducer principles, such as gravimetric viral diagnostics and more,would enhance the manuscript. But it's not a must, only a suggestion.

Author Response
The manuscript presents a comprehensive and systematic study on a number of biosensors for avian influenca virus based on nanobio hybrid materials. The text is clearly written. The style is fine, but minor spell check is required. This relates to the lines 36, 60, 61, 70, 103, 121, 134, 140, 150, 153, 195, 209, 214, 229, 232, 277, 299, 306, 315, 321, 336, and 338. Maybe few more.
Authors agreed with reviewer’s suggestion, we modified the mistake with thre red color.
In addition, at the end of the manuscript, a short view on competing transducer principles, such as gravimetric viral diagnostics and more,would enhance the manuscript. But it's not a must, only a suggestion.
Authors agreed with reviewer’s suggestion, we descriped the gravimetric viral diagnostics. Please check reference 84. Thanks for reveiwer's comment that improve the manuscript quaility.
Round 2
Reviewer 1 Report
One concern is that this review is not very comprehensive, the authors missed one big area in IV detection:the magnetic sensors. Line 94-96. Inadequate references on the previous work in IV detection methods. There are many groups exploring the magnetic nanosensors for the detection of IVs. For example: Krishna, Venkatramana D., et al. "Nanotechnology: Review of concepts and potential application of sensing platforms in food safety." Food microbiology 75 (2018): 47-54. Wu, Kai, et al. "Portable GMR Handheld Platform for the Detection of Influenza A Virus." ACS sensors 2.11 (2017): 1594-1601.
Author Response
One concern is that this review is not very comprehensive, the authors missed one big area in IV detection:the magnetic sensors. Line 94-96. Inadequate references on the previous work in IV detection methods. There are many groups exploring the magnetic nanosensors for the detection of IVs. For example: Krishna, Venkatramana D., et al. "Nanotechnology: Review of concepts and potential application of sensing platforms in food safety." Food microbiology 75 (2018): 47-54. Wu, Kai, et al. "Portable GMR Handheld Platform for the Detection of Influenza A Virus." ACS sensors 2.11 (2017): 1594-1601.
Authors agreed with reviewer’s concern. We added the magnetic-based AIV detection method. Please check the reference 25, 33, 85-88 with blue color.
Reviewer 2 Report
I may recommend this revised version of the paper for publication in the Micromachines.
Author Response
Authors appreicated reviewer's comments for the manuscripts . Those are meaningful for better manuscript.
Reviewer 3 Report
The work is well organized and comprehensively described. In addition, more appropriate and adequate references to related and previous work had been added. English language and style are fine for me. The work is now worth to be published in te present form.
Author Response

(The authors gave the same response as above.)
